# Traveling wave of inflammatory response to regulate the expansion or shrinkage of skin erythema

**Maki Sudo, Koichi Fujimoto** *

Department of Biological Sciences, Graduate School of Science, Osaka University, Machikaneyama-cho, Toyonaka, Japan

* fujimoto@bio.sci.osaka-u.ac.jp

## Abstract

Many skin diseases show circular red lesions on the skin, called erythema. Erythema is characterized by the expansion of its circular area solely from local stimulation. A pathological inflammatory response caused by the stimulation persistently increases inflammatory mediators in the dermis, whereas a normal inflammatory response transiently increases mediators, resulting in the shrinkage of the erythema. Although the diffusion of mediators theoretically reproduces the expansion, how the inflammatory response expands or shrinks the erythema remains unknown. A possibility is positive feedback, which affects mediator production and can generate two distinct stable states (i.e., inflamed and noninflamed), referred to as bistability. Bistability causes a state transition either from the noninflamed to inflamed state or the reverse direction by suprathreshold stimulation. Additionally, the diffusion selectively causes state transition in either direction, resulting in spatial spread of the transited state, known as the traveling wave. Therefore, we hypothesize that the traveling wave of the inflammatory response can account for both the expansion and shrinkage. Using a reaction-diffusion model with bistability, we theoretically show a possible mechanism in which the circular inflamed area expands via the traveling wave from the noninflamed to the inflamed state. During the expansion, the boundary between the inflamed and noninflamed areas moves at a constant velocity while maintaining its concentration gradient. Moreover, when the positive feedback is weak, the traveling wave selectively occurs from the inflamed to noninflamed state, shrinking the inflamed area. Whether the inflamed area expands or shrinks is mainly controlled by the balance of mediator concentration between the noninflamed and inflamed states, relative to the threshold. The traveling wave of the inflammatory response provides an experimentally testable framework for erythema expansion and shrinkage, thereby contributing to the development of effective treatments, including probiotics.

## Background

Numerous inflammatory skin diseases, including eczema, urticaria, psoriasis, infectious diseases, and lymphomas, lead to circular red lesion areas on the skin called erythema [1, 2].

**Data Availability Statement:** All relevant data are within the manuscript and its Supporting information files.

**Funding:** This work was supported by Grants-in-Aid for Scientific Research from the Ministry of

Education, Culture, Sports, Science and Technology of Japan to KF (17H06386). The funders had no role in study design, data collection and analysis, decision to publish, or preparation of the manuscript.

Erythema is caused by various pathogenic factors, such as physical stimulation, chemical drugs, and bacterial infections (Fig 1Ai) [1]. These factors lead to inflammatory response, that is, the secretion of inflammatory mediators, such as cytokines (e.g., TNF-$\alpha$ and IL-1$\beta$) and histamine (Fig 1Aii) [3–5]. These mediators increase blood volume in the dermal blood vessels (Fig 1Aiii) [6, 7]. The increase in the blood volume appears as erythema (Fig 1Aiv), which typically appears over a few millimeters solely by local transient stimulation (e.g., in a few minutes) [8] and expands to a few centimeters in a few days [1, 8]. During expansion, the lesions are well-circumscribed (i.e., thick red-colored) in some diseases or skin conditions (Fig 1B) and poorly circumscribed (i.e., light red-colored) in others (Fig 1C) [9]. The expansion of multiple erythemas leads to their fusion [1]. Autonomous expansion is an indispensable event during disease progression. In a pathological case, inflammatory response persists, and concentrations of mediators fail to return to the original levels [10–13]. In contrast, the inflammatory response in the healthy skin (normal inflammatory response) initiates a temporal increase in the level of mediators, which returns to original levels [10–13]. Controlling the inflammatory response to suppress the expansion velocity and further shrink the erythema can offer indications for developing effective treatments. Nevertheless, how the inflammatory response controls the expansion and shrinkage of erythema remains unclear because of the experimental difficulty in detecting spatiotemporal dynamics of inflammatory mediators in the dermis.

In recent years, mathematical models and computer simulations have predicted the spatiotemporal dynamics in the dermis [14–19]. One model incorporated experimentally known biochemical or transcriptional regulation of mediators and their intercellular diffusion and highlighted the expansion of a well-circumscribed lesion [14]. This type of model incorporates the reaction and diffusion of molecules, referred to as the reaction–diffusion model [20, 21]. Another reaction–diffusion model incorporating self-activation, that is, positive feedback of histamine, has also shown expansion [15]. These models show that the diffusion of mediators could cause erythema expansion. The common regulation of mediators responsible for the expansion in these two models is the positive feedback. Experiments support the positive feedback in the dermis, that is, the inflammatory mediators activate NF-$\kappa$B signaling, and their production is induced in response to NF-$\kappa$B activation [22]. Mathematically, positive feedback can generate two distinct and stable states (i.e., inflamed and noninflamed states) called bistability. Bistability causes a persistent transition between the noninflamed and inflamed states by a suprathreshold stimulation [13]. Diffusion and bistability selectively cause transition from one (e.g., noninflamed) state to another (e.g., inflamed), resulting in the spatial spread of the state transitions, referred to as the traveling wave [20, 23]. Furthermore, weak positive feedback selectively causes a reverse transition, e.g., from inflamed state to noninflamed state, resulting in a traveling wave in the reverse direction [20, 23]. Therefore, we hypothesized that the diffusion and bistability of inflammatory mediators could account for both expansion and shrinkage.

This study develops a bistable reaction–diffusion model to determine whether and how diffusion and bistability can cause expansion and shrinkage. The expansion of the inflamed area appears as the traveling wave of the transition from the noninflamed to inflamed state. We further demonstrate that diffusion and bistability can shrink the inflamed area through a traveling wave of a reverse transition from the inflamed to noninflamed state, depending on the strength of the positive feedback.

## Methods

To analyze the expansion of erythema, we formulated a reaction–diffusion model [20, 21, 23]. Because mediator-secreting cells (e.g., immune cells and keratinocytes) do not exhibit spatial

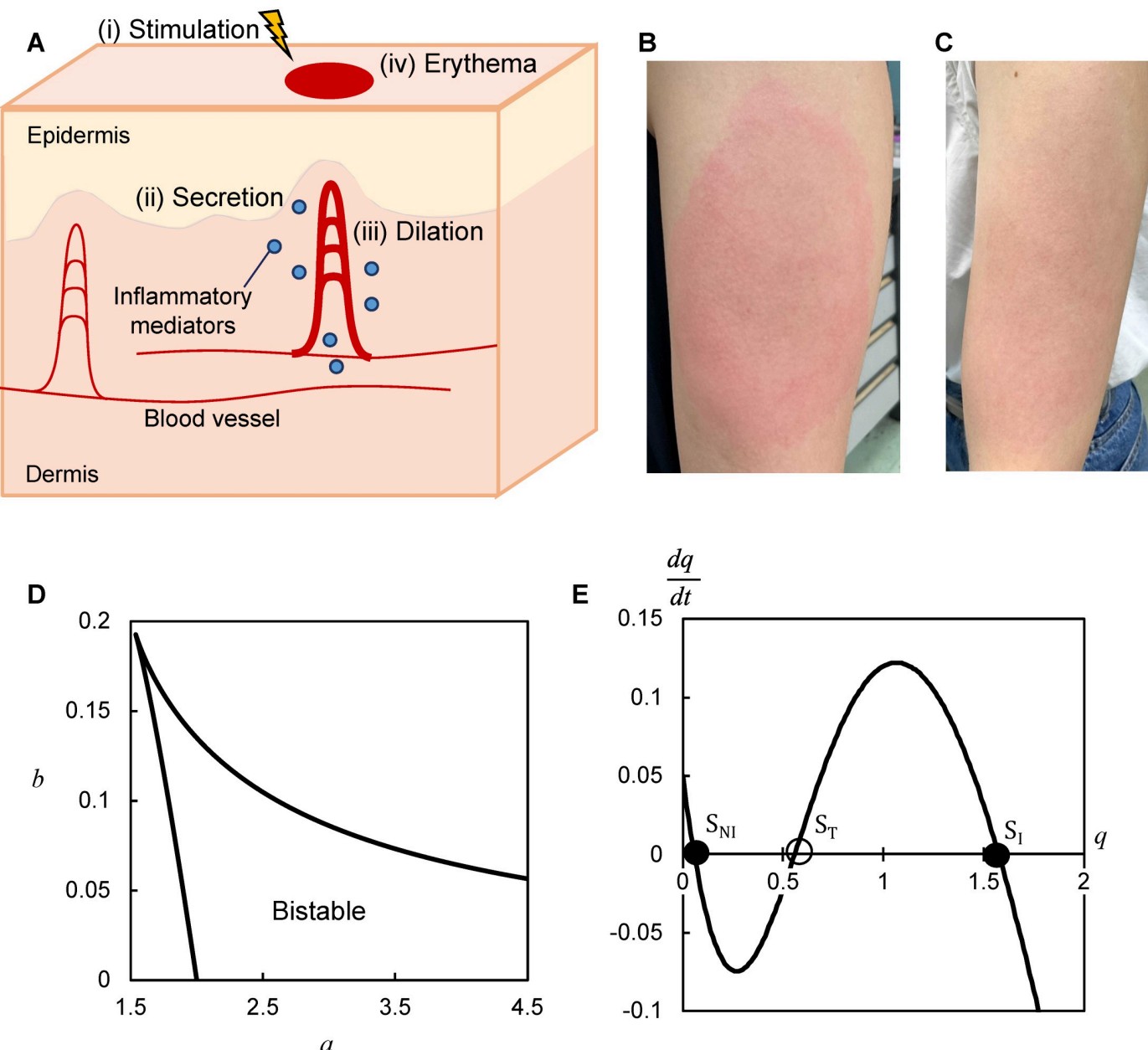

**Fig 1. Modeling of the expansion of erythema. (A)** Process of the inflammatory response for erythema development. When keratinocytes in the epidermis and resident immune cells in the dermis are stimulated (i), they secrete inflammatory mediators that induce their own production from these mediator-secreting cells (ii). The mediators diffuse in the dermis and cause the dilation of local blood vessels (iii). The dilation appears as redness on the skin surface, forming erythema (iv). **(B, C)** Photographs of erythema expansion showing well-circumscribed lesion **(B)** and poorly circumscribed lesion **(C)** of mRNA COVID-19 vaccine. **(D)** Range (surrounded by black solid line) of $a$ and $b$ such that the model (Eq 3, $d = 0$) exhibits bistability. **(E)** Kinetics of bistability in Eq (3) represented by the production rate of mediators ($\frac{dq}{dt}$) as a function of the concentration ($q$). Two filled circles represent stable steady states ($S_{NI}$, $S_I$), whereas the hollow circle indicates an unstable steady state ($S_T$). This system can switch between the stable states of low ($S_{NI}$)- and high ($S_I$)- concentration depending on the perturbation, such as initial stimulation or diffused mediators ($a = 2.14$, $b = 0.05$).

localization in the dermis [1, 4, 24], we assumed a homogeneous distribution of these cells in the two-dimensional space of the dermis along the skin surface (Fig 1A). We first formulated the following equation by introducing the observed biochemical or transcriptional regulation of the inflammatory mediator's concentration ($p$) in the intracellular and extracellular

environments into an ordinary differential equation:

$$\frac{dp}{dT} = \frac{\alpha p^n}{p^n + K_M{}^n} + \beta - \gamma p. \tag{1}$$

The first, second, and third terms represent the induction of own production (i.e., the positive feedback) [22], basal secretion [25], and degradation [26], respectively. Here, $\alpha$, $n$, $K_M$, $\beta$, and $\gamma$ denote the maximum production rate, Hill coefficient of the cooperativity, threshold of production, basal secretion rate, and degradation rate, respectively. The values of these parameters can depend on the skin condition. For example, experiments have suggested that the maximum production rate ($\alpha$) of one type of mediator, IL-1$\beta$, increased with the deterioration of skin microbiome [27], and that the basal secretion rate ($\beta$) of IL-1$\beta$ increased with the deficiency of the skin barrier integrity [25].

Then, we introduced the diffusion to formulate a reaction–diffusion equation:

$$\frac{\partial p}{\partial T} = \frac{\alpha p^n}{p^n + K_M{}^n} + \beta - \gamma p + D\Delta p, \tag{2}$$

where $D$ and $\Delta$ in the fourth term denote the diffusion coefficient and the Laplacian operator $\left(\frac{\partial^2}{\partial x^2} + \frac{\partial^2}{\partial y^2}\right)$, respectively [28]. Eq (2) becomes identical with the previous model incorporating both inflammatory mediator and its substrate as variables [14] when the substrate is assumed to be in a steady state (See Appendix A2 in S1 Appendix for a detailed derivation). We set $n$ (Eq 2) to 2 to introduce the simplest form of the cooperativity required for the bistability. Because there is no quantitative information on the other kinetic parameter values ($\alpha$, $\beta$, $\gamma$, and $K_M$) and diffusion coefficient ($D$), we investigated the model dynamics for a wide range of parameters, such that the model exhibits bistability. For this purpose, we non-dimensionalized Eq (2) by normalizing the variables and parameters as follows (See Appendix A1 in S1 Appendix for a detailed derivation):

$$\frac{\partial q}{\partial t} = \frac{aq^n}{q^n + 1} + b - q + d\Delta q, \tag{3}$$

where $q$, $t$, $a$, $b$, and $d$ are the normalized concentration of mediator, normalized time, normalized maximum production rate, normalized basal secretion rate, and normalized diffusion coefficient, respectively.

We analytically determined the range of $a$ and $b$ such that the model exhibits bistability in the absence of diffusion (Fig 1D); there are two stable states given by low and high concentrations, corresponding to the noninflamed ($S_{NI}$ in Fig 1E) and inflamed ($S_I$ in Fig 1E) state, respectively. We assumed that an area with a concentration of $S_I$ in the dermis appears as erythema on the skin surface. In this setting, the model has one unstable steady state, corresponding to a threshold concentration ($S_T$ in Fig 1E) for the transition between the two stable states. $S_{NI}$ is maintained for a subthreshold perturbation (i.e., below the concentration of $S_T$), whereas it transits to $S_I$ for a suprathreshold perturbation (i.e., above the concentration of $S_T$). $S_I$ also transits to $S_{NI}$ when the concentration decreases below $S_T$.

Finally, as an initial condition of the model simulation (Eq 3), we referred to the physiological condition at the onset of erythema, where one or a few small (~1mm) inflamed areas exhibited a concentration of mediators above the threshold ($S_T$) [1]. In contrast, the surrounding areas exhibited a concentration of mediators below the threshold in the dermis [1]. Based on these observations, for each inflamed area, we set a circular area of $q > S_T$ and the surrounding area of $q < S_T$, which are given by the two-dimensional Gaussian distribution (Fig 2A, $t = 0$). Given this initial condition, numerical simulation of Eq (3) was performed in two-dimensional

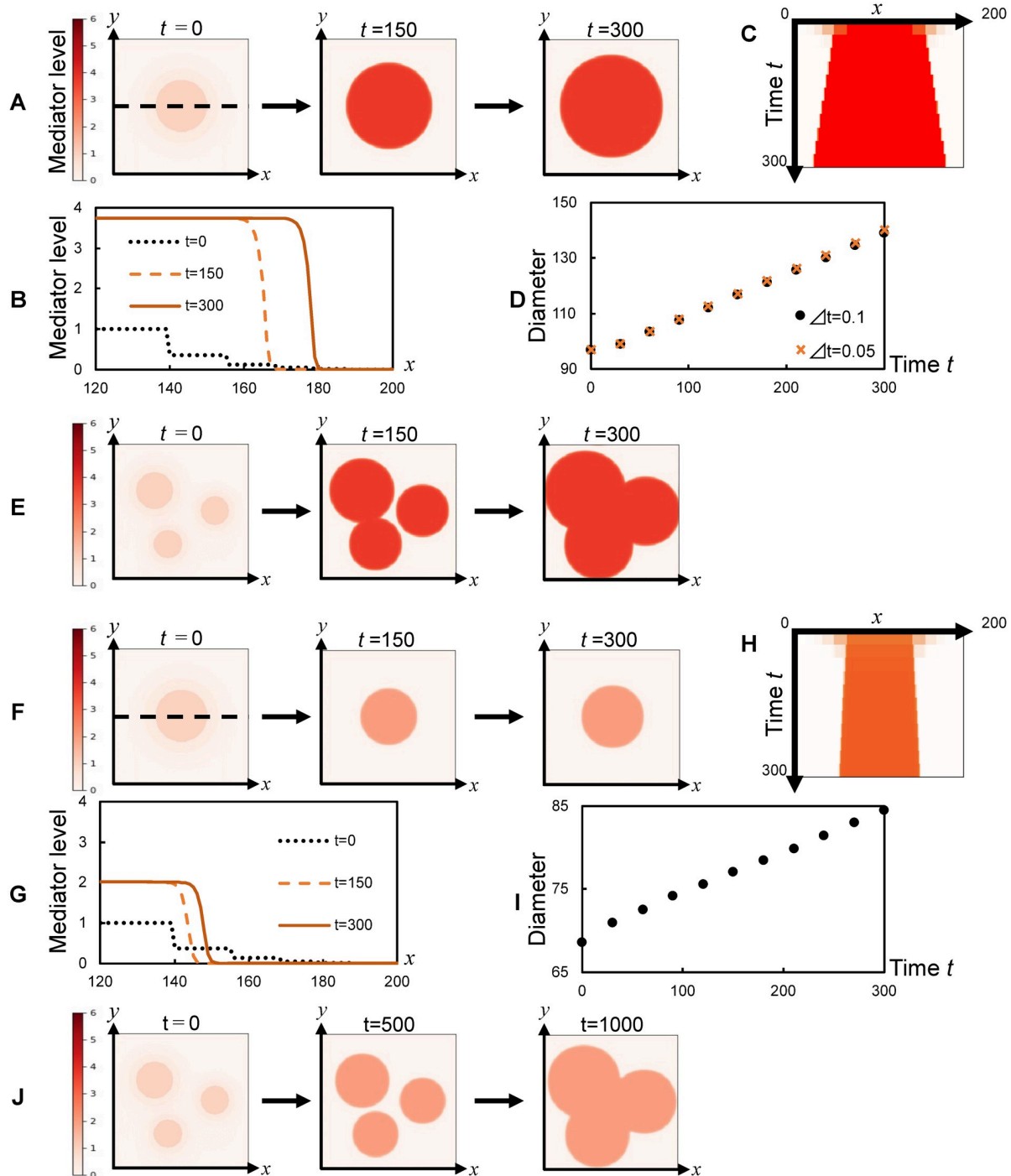

**Fig 2. Diffusion and bistability can cause expansion via the traveling wave. (A, E, F, J)** Spatiotemporal evolution of inflammatory mediator levels ($q$; inset at the left) upon initial stimulation in a circular area (**A**; $a$ = 4.0 and **F**; $a$ = 2.5) and in three separate areas (**E**; $a$ = 4.0 and **J**; $a$ = 2.5). **(B, G)** Spatial pattern of mediator levels at three different time points, **(C, H)** spatiotemporal evolution in the inflamed area, and **(D, I)** temporal evolution of the diameter of the inflamed area (above the unstable steady state $S_T$, $q$ = 0.26 for **D**, $q$ = 0.69 for **I**; red), at $y$ = 100 in **A** (**B–D**) and **F** (**G–I**) (dashed line in the left panel). $b$ = 0.01, $d$ = 0.5 in **A–J**. The obtained results were almost the same for $\Delta t$ = 0.05 and $\Delta t$ = 0.1 in **D**.

geometry under periodic boundary conditions using a finite difference scheme of the first-order approximation in time and space, referred to as the Euler method:

$$\frac{q(t + \Delta t, x, y) - q(t, x, y)}{\Delta t}$$
$$= \frac{aq(t, x, y)^n}{q(t, x, y)^n + 1} + b - q(t, x, y) + d\left(\frac{q(t, x + \Delta x, y) + q(t, x - \Delta x, y) - 2q(t, x, y)}{\Delta x^2} + \frac{q(t, x, y + \Delta y) + q(t, x, y - \Delta y) - 2q(t, x, y)}{\Delta y^2}\right),$$

where $\Delta t$, $\Delta x$, and $\Delta y$ were chosen to satisfy Von Neuman stability. We confirmed that the obtained results were barely influenced by the choice of the temporal discretion size $\Delta t$ (Fig 2D). A simulation code written in C language is available from GitHub: https://github.com/MakiSudo/Travelingwave_Simulation/blob/bc2c10ddd5eff8db374b0804e11a63ef3c0e766a/Simulationcode.c.

## Results

### Diffusion and bistability can cause expansion of circular inflamed area

We examined whether diffusion and bistability can cause expansion of the erythema in the model. The model simulations showed that a circular inflamed area was initially caused by a transient and local perturbation to the mediator's concentration and subsequently expanded centrifugally over time (Fig 2A), consistently with the expansion of erythema (Fig 1B). During the expansion, the inflamed area maintained a steep gradient of concentration at the boundary (Fig 2B) and increased the diameter at a constant rate (velocity) over time (Fig 2C and 2D). The mediator level was persistently high (S1A and S1B Fig in S1 Appendix), consistently with the pathological inflammatory response [13]. This model further reproduced the fusion of multiple inflamed areas (Fig 2E). Even when the ratio of the inflamed and noninflamed concentration ($S_I$ /$S_{NI}$) was smaller according to a decrease in the maximum production rate ($a$), the inflamed area similarly expanded (Fig 2F) with a steep boundary gradient (Fig 2G) at a constant velocity (Fig 2H and 2I) and fused (Fig 2J). We then examined whether diffusion is necessary for expansion. Without diffusion (i.e., $d = 0$), an inflamed area appeared; however, this area did not expand and remained constant over time (S1C Fig in S1 Appendix). We next examined whether bistability is necessary for expansion. When bistability was lost by further decreasing $a$, a local transient stimulation caused an inflamed area, but the area did not expand (S1D Fig in S1 Appendix). Similar results were obtained when the bistability was lost by decreasing the basal secretion rate ($b$) (S1E Fig in S1 Appendix). Thus, these results confirmed that diffusion and bistability could cause the expansion of the circular inflamed area.

### Expansion caused by a traveling wave of the transition from a noninflamed to inflamed state

The expansion follows spatiotemporal changes in inflammatory mediator concentration. First, an initial suprathreshold perturbation locally induces an inflamed area (Fig 2A, 2B, 2F and 2G, time = 0). In this area, the production of mediators increases by positive feedback. The produced mediators diffuse to the adjacent noninflamed area. In the noninflamed area, the diffused mediators are large enough to become a suprathreshold perturbation, causing a selective transition from the noninflamed state ($S_{NI}$) to the inflamed state ($S_I$) at the boundary between the inflamed area and the noninflamed area (Fig 2A, 2B, 2F and 2G, e.g., time = 150). This series of events, that is, positive feedback of production, diffusion, and state transition in the adjacent area, occurs in each position and propagates to the surrounding noninflamed area. Thus, the inflamed area expands as the traveling wave, while maintaining the velocity (Fig 2D and 2I) and a gradient at the boundary (Fig 2B and 2G). Therefore, diffusion and bistability

cause the traveling wave of selective transition from the noninflamed to the inflamed state, resulting in the expansion.

## Control of bistability or diffusion to suppress the expansion velocity

We examined whether the expansion velocity of the inflamed area can be suppressed by controlling diffusion and bistability. The expansion velocity monotonically decreased with a decrease in the diffusion coefficient ($d$) (Fig 3A). It also decreased with a decrease in the maximum production rate ($a$) or basal secretion rate ($b$) controlling the positive feedback activity (Fig 3B–3E and S2A-S2D Fig in S1 Appendix). Unlike the dependence on $d$, the velocity continuously decreased and fell below zero at a threshold value of $a$ and $b$ (Fig 3B, 3C and 3F). Note that the parameter value did not affect the gradient (Fig 3G–3I). Thus, these results show that the expansion velocity was suppressed by diffusion and positive feedback activity for a given bistability.

## Erythema shrinkage by controlling bistability

Below the threshold value of $a$ or $b$, the expansion velocity became negative, where transition selectively occurred from the inflamed ($S_I$) to the noninflamed ($S_{NI}$) state (Fig 3B, 3C and 3F). This traveling wave resulted in the shrinkage of the inflamed area (Fig 3B, 3C, 3J and S2E–S2G Fig in S1 Appendix). During the shrinkage, the mediator level at the initial stimulation transiently increased to the inflamed state and then returned to the original noninflamed state (S2H Fig in S1 Appendix), consistently with the normal inflammatory response [13]. Therefore, these results showed that diffusion and bistability accounted for the normal inflammatory response leading to the shrinkage as well as the pathological response leading to the expansion.

## Balance between the inflamed and noninflamed state concentrations determines expansion or shrinkage

Finally, we clarify how bistability controls the expansion and shrinkage (Fig 4A). To theoretically formulate the velocity of the traveling wave using the model parameters [20], we approximated Eq (3) to

$$\frac{\partial q}{\partial t} = A(q - S_{NI})(S_T - q)(q - S_I) + d\Delta q, \tag{4}$$

under an assumption that $\frac{1}{S_T{}^n + 1}$ is approximated to a constant $A$ (See Appendix A3 in S1 Appendix). Mathematically, the velocity is approximately determined by the diffusion coefficient ($d$) and the concentrations at the three steady states (i.e., $S_{NI}$, $S_T$, $S_I$):

$$v = \sqrt{\frac{Ad}{2}}(S_{NI} + S_I - 2S_T). \tag{5}$$

The theoretical velocity is proportional to the square root of the diffusion coefficient ($d$), and more interestingly, to the difference between the concentrations of the stable ($S_{NI}$ and $S_I$) relative to unstable ($S_T$) states. For the range of parameter values showing bistability, the theoretical velocity agreed with the simulated velocity in the sign (Figs 3F and 4A) and the approximate value, except for the parameter values at the velocity of zero (S3 Fig in S1 Appendix). When $S_T$ is closer to $S_{NI}$ than to $S_I$, the velocity is positive, indicating the expansion of the inflamed area (Fig 4A and 4B). When $S_T$ is at an equal distance from $S_{NI}$ and $S_I$, given a decrease in the maximum production rate ($a$), the velocity is suppressed toward zero (Fig 4A

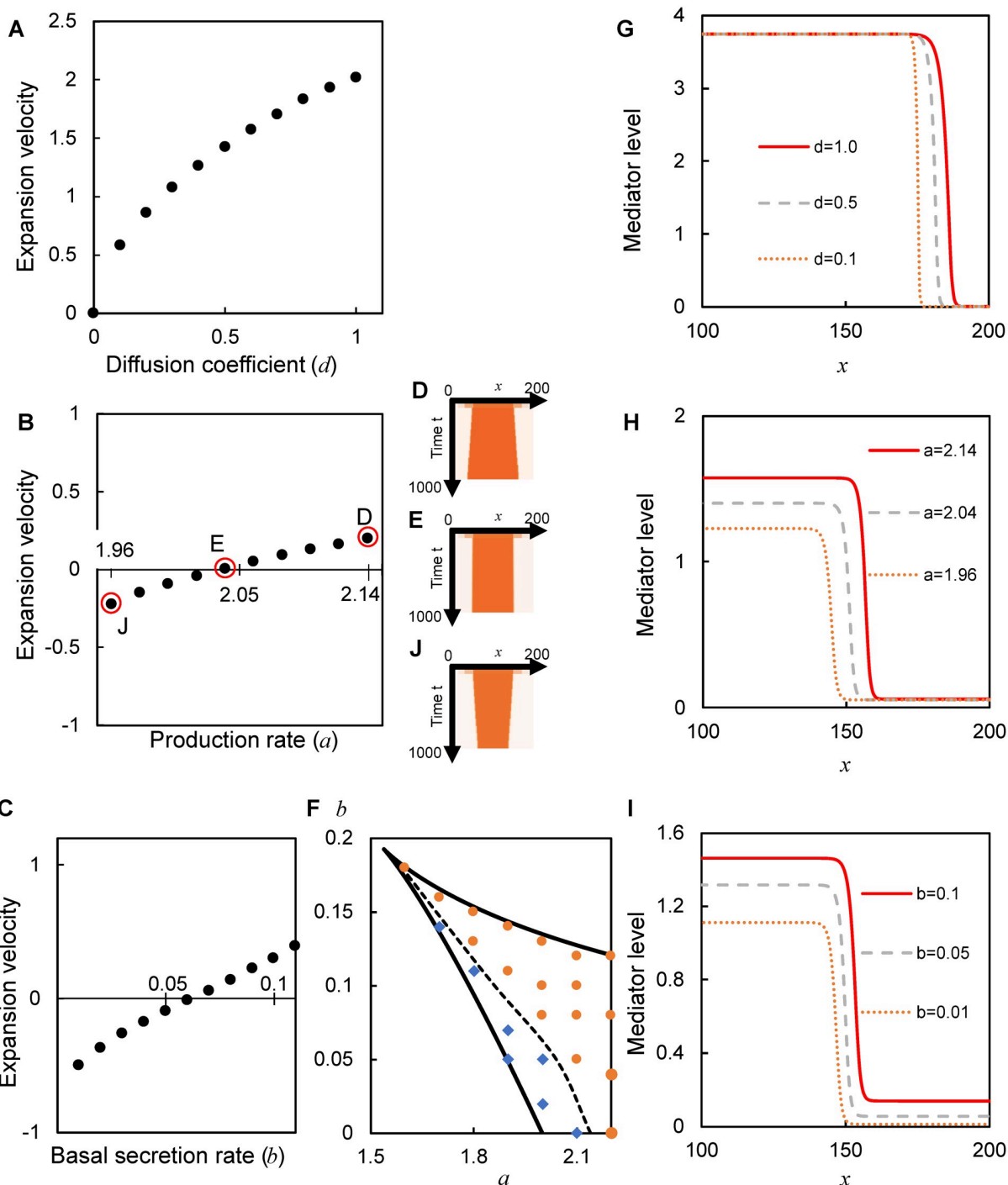

**Fig 3. Expansion velocity is controlled by diffusion and positive feedback for given bistability. (A–C)** Dependence of the expansion velocity on the diffusion coefficient ($d$; **A**), maximum production rate ($a$; **B**), and basal secretion rate ($b$; **C**). (**D, E, J**) Spatiotemporal changes in the inflamed area, which is above the unstable steady state $S_T$ (**D**) $q = 0.55$; © $q = 0.63$; (**J**) $q = 0.72$, respectively, for three different values of maximum production rate indicated in **B**. (**F**) Simulation results were superimposed on the theoretically calculated range of bistability (surrounded by black solid line) shown in Fig 1D. Symbols represent the expansion (orange circles) and shrinkage (blue diamonds). The dashed line represents the theoretical velocity of zero ($v = 0$) calculated from Eq 5. $d = 0.5$. (**G–I**) Dependence of the spatial pattern on the diffusion coefficient ($d$; **G**), maximum production rate ($a$; **H**), and basal secretion rate ($b$; **I**). $a = 4$, $b = 0.01$ in **A, G**. $b = 0.05$, $d = 0.5$ in **B, D, E, H, J**. $a = 2$, $d = 0.5$ in **C, I**.

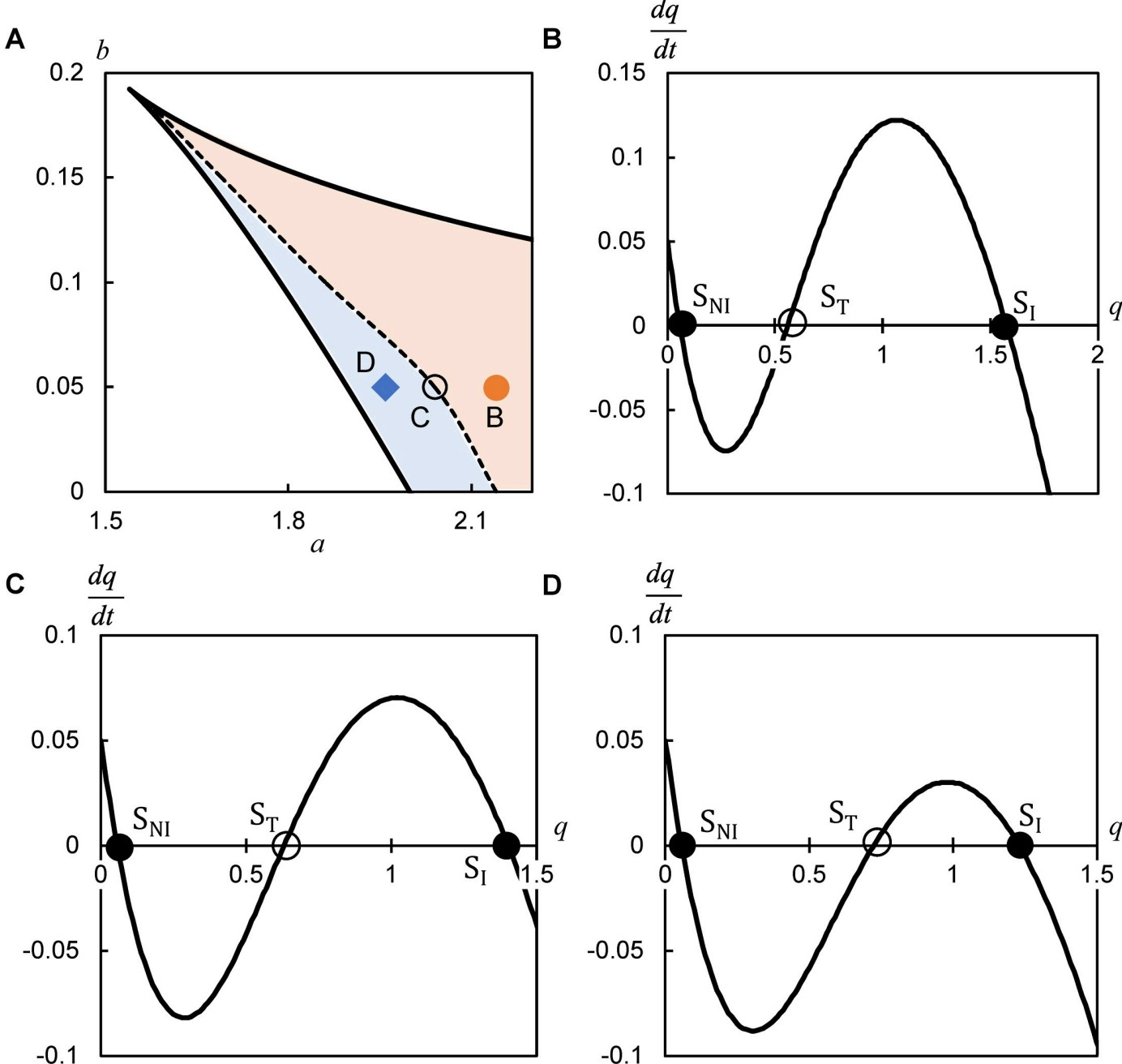

**Fig 4. Balance of mediator concentration regulates expansion or shrinkage.** (A) Range of $a$ and $b$ for the velocity $v > 0$ (calculated from Eq 5), indicating the expansion (orange), $v < 0$, indicating the shrinkage (light blue), and $v = 0$ (dashed line). $d = 0.5$. (B–D) Kinetics of bistability in Eq (3) represented by the production rate of mediators $\left(\frac{dq}{dt}\right)$ as a function of the concentration ($q$) for $a = 2.14$ (B), $a = 2.04$ (C), $a = 1.96$ (D). $b = 0.05$ in B–D. Two filled circles ($S_{NI}$, $S_I$) represent stable steady states, whereas the hollow circle ($S_T$) indicates an unstable steady state.

and 4C). The velocity is negative, indicating the shrinkage of the inflamed area when $S_T$ is closer to $S_I$ than to $S_{NI}$ (Fig 4A and 4D). Similar results were obtained by decreasing the basal secretion rate ($b$). Therefore, depending on whether the threshold concentration ($S_T$) is closer to the noninflamed state ($S_{NI}$) or inflamed state ($S_I$), erythema expands or shrinks, respectively

(Fig 5). The balance of mediator concentrations (i.e., how close the threshold is to the noninflamed or inflamed state) further determines the velocity of expansion or shrinkage.

## Discussion

### Diffusion and bistability cause both expansion and shrinkage of erythema as the traveling wave

Erythema is characterized by the expansion of its circular area. In pathological skin showing expansion, inflammatory response persists and the concentration of mediators fails to return to the original level [10–13]. Recent theoretical studies have shown two independent mechanisms of how the pathological inflammatory response causes expansion [14, 15]. One is the diffusion of inflammatory mediators and the other is positive feedback, which can generate bistability. However, the mechanism of how diffusion and bistability can cause expansion remains unknown. Furthermore, how we can reduce the expansion velocity and further shrink the erythema has not yet been identified. In this study, we theoretically show that diffusion and bistability can synergistically cause not only expansion (Fig 2) with the pathological inflammatory response (S1B Fig in S1 Appendix) but also shrinkage (Fig 3; S2E–S2G Fig in S1 Appendix) with the normal inflammatory response (S2H Fig in S1 Appendix) by the traveling wave. Whether the inflamed area expands or shrinks is determined by the concentration balance between the noninflamed ($S_{NI}$) and inflamed ($S_I$) state relative to the threshold ($S_T$; Eq 5). Expansion occurs when $S_T$ is closer to $S_{NI}$ than to $S_I$, whereas shrinkage occurs when $S_T$ is closer to $S_I$ than to $S_{NI}$ in a wide range of parameters controlling the inflammatory response, including positive feedback, basal secretion, and degradation (Fig 4; S3 Fig in S1 Appendix). An interesting future study would be the analysis of whether the balance captures such inflammatory wave dynamics in other bistable systems, with more complex biochemical reactions [12, 14, 15]. Therefore, the balance of bistable states could provide an experimentally testable framework for the normal and pathological inflammatory responses (Fig 5).

### Expansion of a well- and poorly-circumscribed erythema

Erythema expands with well-circumscribed lesions (Fig 1B) or with poorly circumscribed lesions (Fig 1C) depending on diseases and skin conditions [9]. A well-circumscribed lesion indicates an inflamed area clearly distinguished from the surrounding noninflamed area [29], which may appear in situations such as a sharp gradient at the boundary and/or a large ratio of the inflamed and noninflamed concentration ($S_I / S_{NI}$) in the dermis. In contrast, a poorly circumscribed erythema, which is difficult to distinguish from the surrounding noninflamed area [29], may appear in situations such as a shallow gradient or a small concentration ratio ($S_I / S_{NI}$) in the dermis. Some of the expansion of inflamed areas in our model appear with the steep gradient and a large ratio of the inflamed and noninflamed concentration ($S_I / S_{NI}$) in the dermis (Fig 2A and 2E), accounting for the possible situations of well-circumscribed lesions. Other expansions appear with a small ratio ($S_I / S_{NI}$) in the dermis (Fig 2F and 2J), consistently with the one possible situation of poorly circumscribed lesions. The consistency can be experimentally verified by checking whether a poorly circumscribed erythema has a small concentration ratio and expands at a constant velocity (e.g., Fig 2I). Alternatively, a well-circumscribed lesion in a deep layer of the dermis may appear to be a poorly circumscribed lesion on the skin surface [29]. Our framework for the expansion of both well- and poorly-circumscribed lesions provides a new perspective on how inflammation in the dermis appears as erythema on the skin surface.

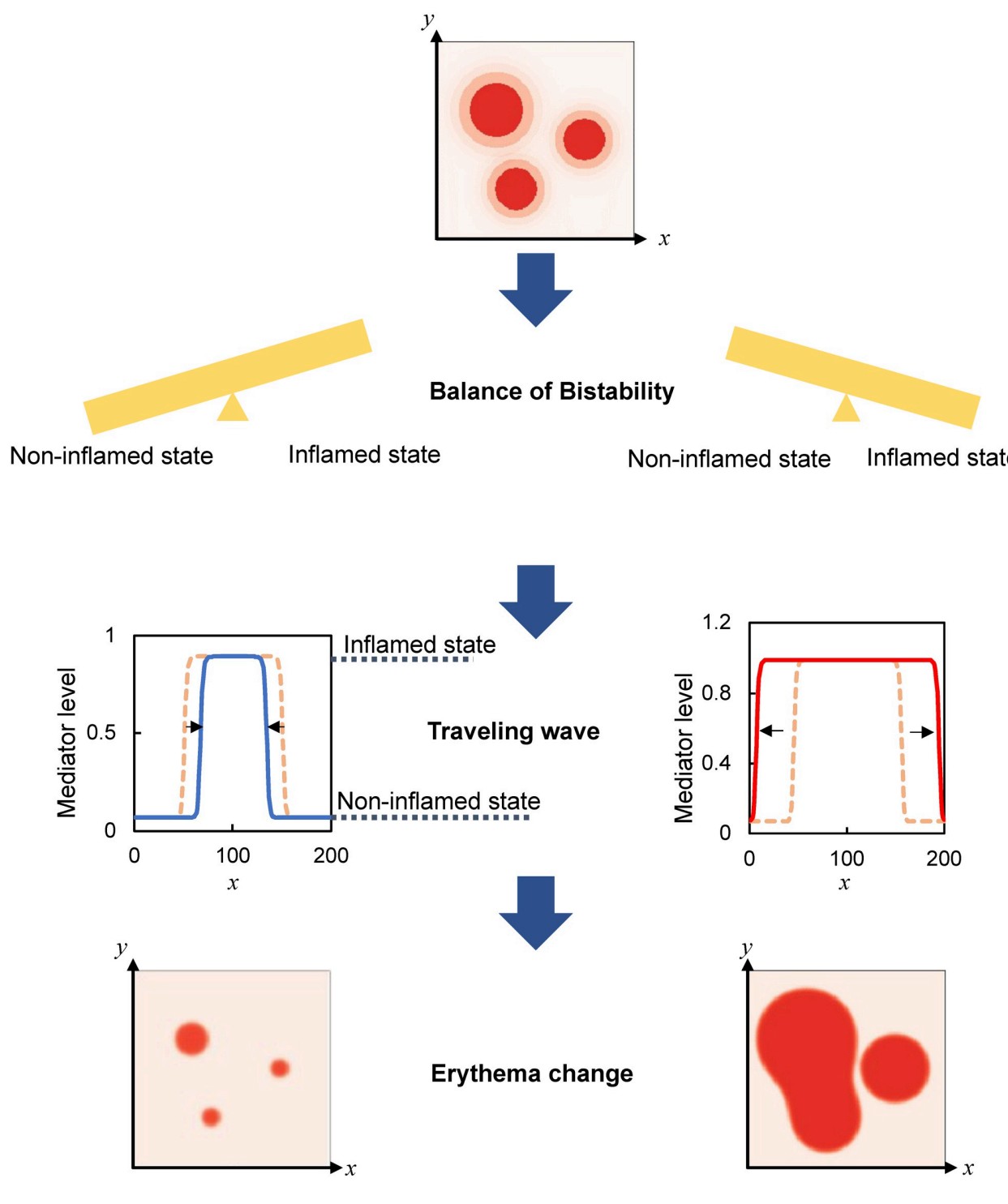

**Fig 5. Traveling wave of inflammatory response regulates the expansion and shrinkage of erythema.**

### Possible relevance to biological factors in the skin

The expansion velocity was suppressed by decreasing the model parameters (Fig 3), which can depend on the skin condition. Experiments have shown that the maximum production rate ($\alpha$ in Eq 2, $a$ in Eq 3) and basal secretion rate ($\beta$ in Eq 2, $b$ in Eq 3) are lower in healthy skin than in pathological skin with a deterioration of the skin microbiome [27] and in skin with deficiency of the skin barrier integrity [25], respectively. Measuring the expansion velocity under different skin conditions will reveal the relation between the model parameters and the skin conditions, thereby potentially providing possible treatments to lower the maximum production rate or the basal secretion rate. For example, probiotics that improve the skin microbiome composition significantly lower the maximum production rate [30]. Additionally, probiotics improve the skin barrier integrity [30], which is expected to lower the basal secretion rate. Thus, probiotics can lower the maximum production rate and the basal secretion rate, thereby possibly reducing the expansion velocity and shrinking erythema. Further study of the relationship between the expansion velocity and the skin condition will offer further insights helpful in developing more effective treatments of erythema.

### Limitation of the present model

In pathological inflammatory response, erythema expands to a certain size and eventually autonomously disappears [1]. Unlike shrinkage in normal inflammatory response (Fig 3J; S2E-S2H Fig in S1 Appendix), the disappearance often shows a decrease in intensity (i.e., redness) without changing the diameter of erythema [14]. Such disappearance could not be reproduced by the present model of inflammatory mediator alone; it may require other factors. Some of the responsible factors are anti-inflammatory mediators, such as IL-10 and TGF-$\beta$ [5, 6]. The anti-inflammatory mediators are produced by inflammatory mediators during the development of erythema, and inhibit the production of inflammatory mediators [22]. A previous mathematical model incorporating the interaction of these mediators accounted for the temporal evolution (i.e., decrease) of the inflammatory response [13], but not the spatiotemporal evolution (i.e., autonomous disappearance). Thus, future studies should extend our model to incorporate the interaction and examine how anti-inflammatory and inflammatory mediators synergistically control the disappearance of erythema.

### Conclusions

In this paper, we demonstrated that diffusion and bistability could cause expansion and shrinkage as a traveling wave. Furthermore, the positive feedback activity of mediator production regulates the transition between the noninflamed and inflamed states, thereby determining whether the inflamed area expands or shrinks (Fig 5). Moreover, regulating the balance of mediator concentration between noninflamed and inflamed states provides an experimentally testable framework for the spatiotemporal evolution of erythema, which can help in the development of effective treatments.

### Supporting information

**S1 Appendix.**
(DOCX)

## Acknowledgments

We thank A. Inoue (Osaka Univ., Japan) for providing clinical photos, and M. S. Kitazawa and K. Matsushita (Osaka Univ., Japan) for stimulating discussion.

## Author Contributions

**Conceptualization:** Maki Sudo, Koichi Fujimoto.

**Formal analysis:** Maki Sudo.

**Investigation:** Maki Sudo.

**Methodology:** Maki Sudo.

**Supervision:** Koichi Fujimoto.

**Writing – original draft:** Maki Sudo, Koichi Fujimoto.

**Writing – review & editing:** Maki Sudo, Koichi Fujimoto.

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
