## [Decision Letter · Decision Letter 0]

28 Apr 2021

PONE-D-21-08023

Traveling wave of inflammatory response to regulate the expansion or shrinking of skin erythema

PLOS ONE

Dear Dr. Fujimoto,

Thank you for submitting your manuscript to PLOS ONE. After careful consideration, we feel that it has merit but does not fully meet PLOS ONE’s publication criteria as it currently stands. Therefore, we invite you to submit a revised version of the manuscript that addresses the points raised during the review process.

We look forward to receiving your revised manuscript.

Kind regards,

Daniele Avitabile

Academic Editor

PLOS ONE

Journal Requirements:

Additional Editor Comments:

Please note that all referees recommend substantial revisions, which I encourage you to consider carefully before deciding on a potential resubmission.

Reviewers' comments:

Reviewer's Responses to Questions

**Comments to the Author**

1. Is the manuscript technically sound, and do the data support the conclusions?

Reviewer #1: Partly

Reviewer #2: Partly

Reviewer #3: Partly

2. Has the statistical analysis been performed appropriately and rigorously? 

Reviewer #1: N/A

Reviewer #2: N/A

Reviewer #3: N/A

3. Have the authors made all data underlying the findings in their manuscript fully available?

Reviewer #1: No

Reviewer #2: Yes

Reviewer #3: No

4. Is the manuscript presented in an intelligible fashion and written in standard English?

Reviewer #1: No

Reviewer #2: Yes

Reviewer #3: Yes

5. Review Comments to the Author

Reviewer #1: This paper developed a bistable reaction-diffusion model that can reproduce expansion and shrinkage of erythema on skin.

Major comments:

Model assumptions, the choice of the model parameter values, and the sensitivity of the results to the parameter values need to be clarified.

Line 74: Why is “a homogeneous distribution of mediator-secreting cells” a biologically reasonable assumption?

Line 80: [21] is with regard to the NfkB pathway. Please explain why this reference can support the positive feedback for “mediators”. A clear definition of “mediators” is needed.

Line 83: Does [24] cover the effects of skin microbiome on all “mediators”?

Line 84: Does [22] cover the effects of barrier damage on all “mediators”?

Line 88: How did you choose the parameters and confirm that their biological plausibility?

Line 100: Why could you choose D=0.5?

Line 106: Why “a few centimeters”, not “a few millimeters” as described in the previous sentence?

Line 137: “decreasing alpha” is not the only way to break bistability. Did you investigate other ways?

Line 152: How much is the result dependent on the choice of the model parameters?

Line 193: Why can you represent the expansion speed by Eq. (3)?

Line 237: “a new control principle” is an overstatement.

Minor comments:

Background is written in an informative manner, but abstract requires a rewrite to tighten up the argument and improve the accessibility. The motivation of the paper needs to be clarified. It was difficult to understand what “other regulations” (l. 15), “diffusion and bistability” (l. 21) and the “balance” (l. 26) mean.

Line 13: Does erythema occur only by transient stimulation, and not by continuous stimulation? What is the definition of “transient”? What is the timescale that the authors consider in this paper?

Line 41: “phenomenon of erythema”: do you man a characteristic feature of erythema?

Line 130: “perturbation” of what?

Line 144: How many is “many”?

Line 201: How wide is “wide”?

Reviewer #2: In this theoretical study the authors developed a bistable reaction-diffusion model to determine whether and how diffusion and bistability cause expansion and shrinkage of skin erythema during inflammatory reaction.

They assumed that expansion of lesion appears as a traveling wave. They showed that diffusion and bistability are necessary to cause expansion.

The paper adds to the new, growing field of modelling of pathological processes in the skin based on reaction-diffusion models. Within the last few years different groups presented models that describe the pattern formation on the skin during urticaria, psoriasis and even skin cancer. Those papers showed how the reaction-diffusion models explain various morphological patterns on the skin such as well demarcated spots, circles and spirals.

1. Although this paper provides yet another approach to modelling of skin inflammation, the message is not clearly presented. It is not clear why the bistable model was chosen. It might be applicable to some situations where erythema (inflammation) is indeed well demarcated, but in many instances this is not the case. For example, in the case of eczema, the lesions are not well demarcated (this is even a defining feature of eczema) and various degrees of inflammation co-exist on the skin surface. In those situations, the bistable model is clearly incorrect. Second, the resolution of erythema only exceptionally happens as shrinkage and most often the intensity of erythema decreases without any changes in the area of inflammation. I would advise the authors to re-think the clinical situations where there model is applicable - I can only think of few such as erysipelas (streptococcal skin infection) and probably urticaria. I would like to know why their model is not generally applicable to any type of inflammation.

2. Please explain the difference between the bistable model used here and the Turing and Scott-Gray reaction-diffusion models used previously. I do not understand in which respect the bistable model could be superior to what has already been modelled.

3. The inherent deficiency of any model of inflammation is lack of measurements confirming the choice of values of the key parameters (such as in Eq. 1). The authors should provide some basis for the choice of the parameters (are they the ones that worked? or was there any support for the choice?)

4. A number of statements and assumptions are simply not true or are not explained. For example, I have no clue why the authors are mentioning the importance of microbiome and skin barrier at all (the is done in several places). Why would secretion rate beta be related to skin barrier integrity?

5. The authors mention that their model is important to model anti-inflammatory treatment but they do not explain why.

Reviewer #3: I have carefully read through this article and checked aspects of the mathematical analysis including the steady states and parameter values for expanding and contracting waves in the model.

The key results are that spread of inflammation called erythema (marked by redness of the skin) could be both expanding or receding, governed by bistable local dynamics.

Overall, I think the article requires major revision because the analysis in parts should be to a higher technical standard, with a clearer exposition of certain key results, relation to and use of well-known theory relevant to bistable reaction-diffusion equations, and correction of inconsistencies (e.g. Fig 4 claims an example of zero expansion speed which cannot be correct according to the theory just mentioned - zero speed requires the integral of the local dynamics to be zero) . I include my annotated version of the article as a PDF upload in order to expand upon these comments.

The study does present the results of primary scientific research, and has not to my knowledge been published elsewhere.

The conclusions are presented in an appropriate fashion and are supported by the data.

The article is presented in an intelligible fashion and is written in standard English.

The research meets all applicable standards for the ethics of experimentation and research integrity.

Regarding data availability, it would be preferable to make simulation code available, or at the very least give a much clearer exposition of the numerical methods used and make sure all parameter values are clearly specific.

6. PLOS authors have the option to publish the peer review history of their article (what does this mean?). If published, this will include your full peer review and any attached files.

Reviewer #1: No

Reviewer #2: No

Reviewer #3: No

---

## [Author Response · Author response to Decision Letter 0]

29 Jun 2021

In response to Reviewer #1: 

>This paper developed a bistable reaction-diffusion model that can reproduce expansion and shrinkage of erythema on skin.

>Major comments: Model assumptions, the choice of the model parameter values, and the sensitivity of the results to the parameter values need to be clarified.

Authors: We thank you for providing your valuable comments on our manuscript. As following the reviewer’s suggestion, we have clarified [A] “model assumptions”, [B] “the choice of the model parameter values”, and [C] “the sensitivity of the results to the parameter values” throughout the manuscript. These revisions are extensively described in our response to your comments on Line 74, 80, 83 and 84 (regarding [A]); Line 88 and 100 (regarding [B]); Line 152 (regarding [C]). Please let us know if there are any other assumptions or results that are not explained.

>Line 74: Why is “a homogeneous distribution of mediator-secreting cells” a biologically reasonable assumption?

Au: This is because these cells do not exhibit a spatial localization in the skin of patients with urticaria [Barzilai A. et al. 2017]. Thus, we assume the homogeneous distribution in the skin. We have incorporated this comment into the relevant sentence (p.4-5 line 81-83; revised manuscript). 

>Line 80: [21] is with regard to the NF-κB pathway. Please explain why this reference can support the positive feedback for “mediators”. A clear definition of “mediators” is needed.

Au: We defined “mediators” as diffusible factors that promote inflammation, such as the pro-inflammatory cytokines IL-1β and TNF-α as described in the previous manuscript (p.3 line 38). This reference [21; Bonizzi G. et al. 2004] shows that IL-1β and TNF-α activate NF-κB, and their production is induced in response to NF-κB activation, supporting the positive feedback for “mediators”. Following the reviewer’s suggestion, we have added the explanation of this positive feedback in Background (p.4 line 62-63; revised manuscript).

>Line 83: Does [24] cover the effects of skin microbiome on all “mediators”? 

>Line 84: Does [22] cover the effects of barrier damage on all “mediators”? 

Au: No, [24; Meisel JS. et al. 2018] reported the effects of skin microbiome only on IL-1β, and [22; Bäsler K. et al. 2017] reported the effects of barrier damage only on IL-1β. To reflect these points, we have revised these texts (p.5 line 91-95; revised manuscript) as follows: “The values of these parameters are expected to depend on the skin condition. For example, the production rate (α) of one type of mediator, IL-1β, has been experimentally suggested to increase with the deterioration of the skin microbiome [29]. The basal secretion rate (β) of IL-1β has been experimentally suggested to increase with a deficiency of the skin barrier protein (ZO-1) [27].”

>Line 88: How did you choose the parameters and confirm that their biological plausibility?

>Line 100: Why could you choose D=0.5?

Au: Because there is no quantitative information on these parameter values including “D”, we now examined the model dynamics for a wide range of all parameters (e.g., new Figures 1D, 3F and 4A) by normalizing these parameters (new Equation 3). We now added this description to the Methods section (p.5-6 line 103-110; revised manuscript). We have selected representative points within the range of parameter values, e.g., the normalized diffusion coefficient d=0.5. We also confirmed that the inflamed area expanded irrespective of d (Fig 3A). Additionally, we showed that the theoretical velocity is proportional to the square root of d (p.11 line 241; revised manuscript). 

>Line 106: Why “a few centimeters”, not “a few millimeters” as described in the previous sentence?

Au: Thank you for correctly pointing out this word. We now corrected the term “centimeters” to “millimeters” (p.6 line 123; revised manuscript).

>Line 137: “decreasing alpha” is not the only way to break bistability. Did you investigate other ways?

Au: Yes, our simulations have confirmed that similar results were obtained when bistability is lost by changing the basal secretion rate (b), as the reviewer may expect. To clarify this result, we now added this description immediately after the relevant sentence (p.8 line 171-172; revised manuscript) and a new figure (S1 Fig).

>Line 152: How much is the result dependent on the choice of the model parameters?

Au: Following the reviewer’s comment, we now analyzed the dependence on the result (i.e., steepness of the mediator concentration gradient) on the parameters, and found that the steepness does not depend much on them in the range of bistability. We added this description in Results (p.10 line 205-206) and a new Figure 3G-I.

>Line 193: Why can you represent the expansion speed by Eq. (3)?

Au: To explain the analytical derivation of the expansion velocity in detail, we now revised Appendix and Results (p.16-17 line 355-375, p.11 line 233-240; revised manuscript).

>Line 237: “a new control principle” is an overstatement.

Au: To support this conclusion, we clarified “the model assumptions”, “the choice of the model parameter values”, and “the sensitivity of the results to the parameter values” throughout the paper, as described above. We now added this point to the corresponding sentence (p.13 line 287-288; revised manuscript) as follows: “Therefore, this balance of bistable states could summarize a new control principle for the normal and pathological inflammatory responses (Fig 5).” 

> Minor comments:

>Background is written in an informative manner, but abstract requires a rewrite to tighten up the argument and improve the accessibility. The motivation of the paper needs to be clarified. It was difficult to understand what “other regulations” (l. 15), “diffusion and bistability” (l. 21) and the “balance” (l. 26) mean.

Au: Following this reviewer's suggestion, we have rewritten the Abstract (p.2; revised manuscript). We hope these revisions improve the accessibility.

>Line 13: Does erythema occur only by transient stimulation, and not by continuous stimulation? What is the definition of “transient”? What is the timescale that the authors consider in this paper?

Au: No, erythema can occur not only by transient stimulation but also by continuous stimulation. Erythema is typically caused by a few minutes of stimulation and lasts few days [Elisabetta Zucchi, et al. 2018; Shimizu 2007]. To clarify this, we now added the description on timescales in Background (p.3 line 40-42; revised manuscript). Because the timescale is not important information in Abstract, we have removed “transient” from Abstract (p.2 line 12; revised manuscript).

>Line 41: “phenomenon of erythema”: do you man a characteristic feature of erythema?

Au: Yes, that is what we meant. Following the reviewer’s comment, we have revised the relevant sentence (p.3 line 40-42; revised manuscript) as follows: “The erythema typically appears in a few millimeters by only local transient stimulation (e.g., in a few minutes) [8] and expands to a few centimeters in a few days [1,8].”

>Line 130: “perturbation” of what?

Au: We have incorporated this comment by adding "to the mediator's concentration" to the relevant sentence (p.8 line157-158; revised manuscript).

>Line 144: How many is “many”?

Au: Our intended meaning of “many” is that the number of diffused mediators is large enough to cause the transition from the noninflamed state to the inflamed state. We have revised these sentences (p.9 line 178-182; revised manuscript) to reflect your comment.

>Line 201: How wide is “wide”?

Au: According to the reviewer’s question, we have confirmed that the theoretical velocities agreed with the simulated velocities for the whole range of parameter values exhibiting the bistability. We have reflected this comment by revising this sentence (p.11-12 line 243-245; revised manuscript) as follows: “For the range of parameter values showing bistability, this theoretical velocity agreed with the simulated velocity in not only the sign (Figs. 3F and 4A) but also the approximate value, except for the parameter values at the velocity of zero (S3 Fig).”

In response to Reviewer #2:

>In this theoretical study the authors developed a bistable reaction-diffusion model to determine whether and how diffusion and bistability cause expansion and shrinkage of skin erythema during inflammatory reaction. They assumed that expansion of lesion appears as a traveling wave. They showed that diffusion and bistability are necessary to cause expansion. The paper adds to the new, growing field of modelling of pathological processes in the skin based on reaction-diffusion models. Within the last few years different groups presented models that describe the pattern formation on the skin during urticaria, psoriasis and even skin cancer. Those papers showed how the reaction-diffusion models explain various morphological patterns on the skin such as well demarcated spots, circles and spirals. 

Authors: We greatly appreciate your valuable concerns on our manuscript. 

>1. Although this paper provides yet another approach to modelling of skin inflammation, the message is not clearly presented. It is not clear why the bistable model was chosen. It might be applicable to some situations where erythema (inflammation) is indeed well demarcated, but in many instances this is not the case. For example, in the case of eczema, the lesions are not well demarcated (this is even a defining feature of eczema) and various degrees of inflammation co-exist on the skin surface. In those situations, the bistable model is clearly incorrect. 

Au: Thank you for your insightful comment. We also think that the bistable model is applicable to the “well-demarcated” (i.e., well circumscribed) erythema, as the reviewer pointed out. Moreover, following the reviewer’s suggestion, we now examined if our model accounts for the expansion of “not well-demarcated” (i.e., poorly circumscribed) erythema, and revised Results (p.8) and Discussion (p.14) as follows: “The poorly circumscribed erythema has been considered to appear in two possible situations depending on the disease [9, 31]. One possible situation is a small difference in the mediator concentration between inflamed (S_I) and noninflamed (S_NI) states, as shown in the present model (Figs 2F-J). The other is the inflammation in a deep layer (e.g., subcutaneous tissue) of the skin [31]. The present bistable model suggests that both situations can give rise to the traveling wave of state transition from noninflamed (S_NI) to inflamed (S_I) state (Fig 2). This prediction can be experimentally verified by checking whether the poorly circumscribed erythema expands at a constant velocity (e.g., Figs 2D and I).” 

However, the present model cannot be applied to another situation raised by the reviewer, where “various degrees of inflammation co-exist on a skin surface.” Reproducing such phenomena might require a spatial heterogeneity of the model parameters such as the basal secretion rate of the mediators, whereas these parameters were set spatially homogeneous across the skin in the present manuscript for the first step of theoretical analysis of the expansion. Thus, simulations of the model with such spatial heterogeneity of the parameter values will not fall in the scope of the current study. 

> Second, the resolution of erythema only exceptionally happens as shrinkage and most often the intensity of erythema decreases without any changes in the area of inflammation. I would advise the authors to re-think the clinical situations where there model is applicable - I can only think of few such as erysipelas (streptococcal skin infection) and probably urticaria. I would like to know why their model is not generally applicable to any type of inflammation.

Au: We agree with your point that the resolution of erythema can be seen after the pathological inflammatory response (i.e., persistent increase of mediator concentration). The present model showed that the normal inflammatory response (i.e., a transient increase in the mediator level) resulting in “shrinkage”, but does not account for the resolution. According to earlier immunological and mathematical studies [Nestle FO et al. 2009, Valeyev NV et al. 2010, Ringham L et al. 2019], the resolution require the anti-inflammatory mediators. Future studies should incorporate the interaction of anti-inflammatory and inflammatory mediators into the present model. We have revised the manuscript to more clearly discuss the resolution (referred to as disappearance in our manuscript) in Discussion entitled “Limitation of the present model” (p.15 line 314-326; revised manuscript).

>2. Please explain the difference between the bistable model used here and the Turing and Scott-Gray reaction-diffusion models used previously. I do not understand in which respect the bistable model could be superior to what has already been modelled.

Au: The Turing model and the Gray-Scott model, unlike our model, consider temporal changes in the inhibitors (such as anti-inflammatory mediators) and in the substrates of inflammatory mediators, respectively. A potential superiority of the present bistable model is the simple formulation, which can be directly derived from the Grey-Scott model (described later) and focuses on the positive feedback. Because of the simplicity, the present model firstly showed both the expansion and shrinkage and clarified the balance between inflamed and noninflamed states as the underlying regulatory principle. These phenomena and principle would be found in the above earlier models. We have added this applicability in Discussion (p.13 line 285-287; revised manuscript). Additionally, the Gray-Scott model equation becomes identical to the bistable model equation, assuming that the substrate concentration reaches an equilibrium state. We now described the relationship between the Gray-Scott and the present bistable model equations in Methods (p.5 line 100-102; revised manuscript) and Appendix (p.16 line 343-353; revised manuscript).

>3. The inherent deficiency of any model of inflammation is lack of measurements confirming the choice of values of the key parameters (such as in Eq. 1). The authors should provide some basis for the choice of the parameters (are they the ones that worked? or was there any support for the choice?)

Au: Because there is no quantitative information on the key parameter values, we now examined the model dynamics for a whole range of all parameters by normalizing the parameters and found that both expansion and shrinkage occurred as long as the bistability exists. The former method was now added to Methods (p.5-6 line 103-110; revised manuscript). As the latter result, we now added new figures (Figure 3B-C, F) and the following sentence to Results (p.11-12 line 243-245; revised manuscript); “For the range of parameter values showing bistability, this theoretical velocity agreed with the simulated velocity in not only the sign (Figs. 3F and 4A) but also the approximate value, except for the parameter values at the velocity of zero (S3 Fig)”. 

>4. A number of statements and assumptions are simply not true or are not explained. For example, I have no clue why the authors are mentioning the importance of microbiome and skin barrier at all (the is done in several places). Why would secretion rate beta be related to skin barrier integrity? 

Au: Following the reviewer’s comment, we have clarified the statements and assumptions on microbiome and skin barrier throughout the manuscript. Regarding “the importance of microbiome and skin barrier” as the reviewer pointed, there is a lot of experimental evidence to support that microbiome and skin barrier affect the production rate (α) and the basal secretion rate (β) of the mediator as described in the previous manuscript (p.5 line 83-85). Additionally, based on the clinical experiments on probiotics and moisturizers [Chen L et al. 2020, Kim BE et al. 2018], we have discussed the possibility that improving the microbiome and skin barrier would decrease α and β, respectively, thereby suppressing the erythema expansion. We have added these discussions to a subsection entitled “Potential application of treatments” in Discussion (p.14 line 299-313; revised manuscript). 

Regarding the final reviewer’s question, we referred to an earlier experimental study [27; Bäsler K et al. 2017] reported that secretion of IL-1β from unstimulated keratinocytes increased upon downregulation of a tight junction protein (ZO-1), which is indispensable for skin barrier integrity. Because this secretion in the unstimulated condition is equivalent to the basal secretion in our model, we assume that “the basal secretion rate (β) is related to the skin barrier integrity”. Taken together, to totally address the reviewer’s 4th comment, we have revised the texts (p.5 line 91–95; revised manuscript) as follows: “The values of these parameters are expected to vary depending on the skin condition. For example, the production rate (α) of one type of mediator, IL-1β, has been experimentally suggested to increase with the deterioration of the skin microbiome [29]. The basal secretion rate (β) of IL-1β has been experimentally suggested to increase with a deficiency of the skin barrier protein (ZO-1) [27].” Please let us know if any other statements and assumptions are not true or are not explained.

>5. The authors mention that their model is important to model anti-inflammatory treatment but they do not explain why.

Au: We meant to discuss the roles of “anti-inflammatory cytokine” but not its “treatment”. Following the reviewer’s suggestion, we have redrafted the relevant sentences in Discussion (p.15 line 314-326; revised manuscript) as follows: “In the pathological inflammatory response, erythema expands to a certain size and eventually autonomously disappears [1]. Unlike the shrinkage in the normal inflammatory response (Fig 3J; S2 Fig E-H), the disappearance often shows a decrease in the intensity (i.e., redness) without changing the diameter of erythema [16]. Such disappearance was not reproduced by the present model of inflammatory mediator alone but may require other factors. One of the responsible factors is anti-inflammatory mediators such as IL-10 and TGF-β [5, 6]. The anti-inflammatory mediators are produced by inflammatory mediators during the development of erythema, and inhibit the production of inflammatory mediators [24]. A previous mathematical model incorporating the interaction of these mediators accounted for the temporal evolution (i.e., decrease) of the inflammatory response, but not the spatiotemporal evolution (i.e., autonomous disappearance). Thus, future studies should extend our model to incorporate the interaction and examine how anti-inflammatory and inflammatory mediators synergistically control the disappearance of erythema.”

In response to Reviewer #3:

>I have carefully read through this article and checked aspects of the mathematical analysis including the steady states and parameter values for expanding and contracting waves in the model. The key results are that spread of inflammation called erythema (marked by redness of the skin) could be both expanding or receding, governed by bistable local dynamics. The study does present the results of primary scientific research, and has not to my knowledge been published elsewhere. The conclusions are presented in an appropriate fashion and are supported by the data. The article is presented in an intelligible fashion and is written in standard English. The research meets all applicable standards for the ethics of experimentation and research integrity. Regarding data availability, it would be preferable to make simulation code available, or at the very least give a much clearer exposition of the numerical methods used and make sure all parameter values are clearly specific. 

Authors: Thank you for your positive evaluation and many insightful comments on our manuscript. We have worked hard to incorporate your feedback, and hope that these revisions satisfy all the comments. Following your suggestion, we have uploaded a simulation code as Supporting Material.

> Overall, I think the article requires major revision because the analysis in parts should be to a higher technical standard, with a clearer exposition of certain key results, relation to and use of well-known theory relevant to bistable reaction-diffusion equations, and correction of inconsistencies (e.g. Fig 4 claims an example of zero expansion speed which cannot be correct according to the theory just mentioned - zero speed requires the integral of the local dynamics to be zero) . I include my annotated version of the article as a PDF upload in order to expand upon these comments. 

Au: We have clarified “the exposition of certain key results” and “relation to and use of well-known theory relevant to bistable reaction-diffusion equations”, and have corrected the inconsistencies throughout the manuscript, as described in the following point-by-point responses to the reviewer’s comments on the manuscript pdf file. 

>Line 15: Not shown to require. A model has shown that diffusible mediators could explain observed propagation. That doesn’t show that such mediators are required.

Au: We now incorporated your comments into Abstract (p.2 line 15-16; revised manuscript) as follows: “Although the diffusion of mediators theoretically reproduces the expansion, how the inflammatory response expands or shrinks the erythema remains unknown.”

>Line 16,17: Maybe “modulation” is more appropriate than “regulation” 

>Line 17: for ?

Au: We have reconstructed the relevant sentence (p.2 line 16-18; revised manuscript) as follows: "A candidate is a positive feedback, which affects mediator production and can generate two distinct stable states (i.e., inflamed and noninflamed), referred to as bistability.”

>Line 24: Steepness depends on parameters? 

Au: Our additional simulations now showed that the steepness of the spatial gradient of mediator concentration did not depend much on parameters as long as the model exhibits bistability. We have added a new Figure 3G-I and the description in Results (p.10 line 205-206).

>Line 27: can be negative. 

Au: We have reconstructed the relevant sentence (p.2 line 26-28; revised manuscript) as follows: “Moreover, as the positive feedback becomes weak given the bistability, the traveling wave selectively occurs from the inflamed to noninflamed state, thereby shrinking the inflamed area.”

>Line 28: What does “the balance” mean? Expansion or shrinkage depends on a well known integral condition and not only the steady state values but also the nonlinearity between them. 

>Line 176: This is well known. Zero speed when integral of the temporal dynamics is zero. 

Au: Our intended meaning of “the balance” is the relative difference of mediator concentration of the inflamed and noninflamed states to the threshold. We agree with the reviewer’s comment that expansion or shrinkage depends on not only the steady state values but also the nonlinearity between the steady states in the model equation, shown by an earlier theory [Volpert V et al. 2009]. Here we approximated that the nonlinear term is a constant (A in Appendix), and theoretically derived the velocity of traveling wave which is proportional to the “balance” (Eq. 5). Even under the assumption, the theoretical velocity showed a similar dependence on the model parameters (i.e., a switch between expansion and shrinkage) to the velocity in the numerical simulations (S3 Fig). Thus, we conclude that the balance is a major component controlling the velocity. Following this comment, we have revised the text (p.2 line 28-30; revised manuscript) as follows: “Whether the inflamed area expands or shrinks and its velocity are approximately controlled by the balance of mediator concentration between the noninflamed and inflamed states, relative to the threshold.”

>Line 31: such as ? 

Au: The candidates are probiotics and moisturizers. Probiotics that improve the skin microbiome composition significantly decreased erythema with decreasing the production rate of the inflammatory mediator [Chen L et al. 2020]. Moisturizers improve the deterioration of the skin barrier. The deterioration increased the basal secretion rate of the inflammatory mediator [Bäsler K et al. 2017]. Based on current experimental findings and our results, probiotics possibly lead to the suppression of expansion and shrinkage of erythema by decreasing the production rate. However, further experimental studies require to clarify the effect of moisturizers on the reduction of basal secretion rate. We have added these discussions to a subsection entitled “Potential application of treatments” in Discussion (p.14 line 299–313; revised manuscript) and added only the probiotics to Abstract (p.2 line 32; revised manuscript).

>Line 50: Why so difficult? Redness should be easy to observe. 

Au: While redness on the skin is easy to observe as the reviewer pointed out, here we intend the experimental difficulty of measuring the spatiotemporal dynamics of mediator concentration in the skin. To avoid such confusion, we have revised the text (p.3 line 51-52; revised manuscript).

>Line 52: Need to give context -there is lots of published relevant mathematical modelling. 

Au: Following the reviewer's suggestion, we reorganized this paragraph. It starts with a review of the previous mathematical models by focusing on the spatiotemporal evolution in skin inflammation and the mathematics of the traveling wave, and finally derives our hypothesis by integrating these previous studies (p.3-4 line 54-72; revised manuscript).

>Line 59: Only “showed” in the context of the assumptions made in those models. This does not show that biologically mediators are required (other mechanisms are not ruled out). 

Au: We have reflected this by replacing “is required for” with “can cause” (p.4 line 60; revised manuscript).

>Line 64: Biologically or mathematically? It is well known mathematically that bistability can lead to forward or reverse waves. 

Au: Following the reviewer’s suggestion, we have thoroughly rewritten these sentences by separately reviewing mathematical studies from biological evidence (p.4 line 66-72; revised manuscript) as follows: “It is known that diffusion, as well as bistability, selectively causes a transition from one (e.g., noninflamed) state to the other (e.g., inflamed) state, resulting in the spatial spread of the state transitions, referred to as the traveling wave [22, 25]. Furthermore, weak positive feedback selectively causes a reverse transition, e.g., from the inflamed state to the noninflamed state, resulting in a traveling wave in the reverse direction [22, 25]. Therefore, we hypothesized that diffusion and bistability of inflammatory mediators can account for not only the expansion but also shrinkage.”

>Line 69: Why necessary.

Au: We have incorporated your comments by replacing “diffusion and bistability are necessary to cause expansion” with “diffusion and bistability can cause expansion” throughout our paper (p.8 line 154, 156, 173, p.9 line 189, p.13 line 276, p.15 line 328; revised manuscript).

> Line 79: Km : usually Kma so p_Km have same units. 

Au: We have reflected this by replacing “K_m” with K_M^n as suggested (p.5 line87, 98 and p.16 line 338; revised manuscript).

>Line 100: Notes exploring the link between this and other models.

Au: Following the reviewer’s suggestion, we added the link between our model and a previous model (Grey-Scott model) in Method & Appendix (p.5 line 100-102; p.16 line 343-353; revised manuscript).

>Line 108: In time? Insufficient detail. How are the diffusion terms approximated?

Au: We introduced a finite difference scheme of the first-order approximation in time and space. To reflect your comment, we have revised this sentence with the formulation of this approximation (p.6 line 127-129). 

>Line 109: How? 

Au: By following this comment, we have changed the sentence (p.7 line 130-131; revised manuscript) as follows: “Δt and Δx were chosen to satisfy Von Neuman stability. We confirmed that the obtained results were not greatly influenced by the choice of the temporal discretion size Δt.”

> Fig 1: Needs some data / real examples. 

Au: Following the reviewer’s comment, we have replaced the illustration in the previous submission with the photographs of erythema (new Figure 1B and C) and got permission on the reuse of these figures from publishers. 

>Line 132: depends on parameters. 

Au: Thank you for the important comment. We found that the steepness of the gradient did not depend much on all the model parameters. We added this description in Results (p.10 line 205-206) and a new Figure 3G-I.

>Line 135: small or zero. 

Au: We have corrected the term “small” to “zero” as the reviewer suggested (p.8 line 168; revised manuscript).

>Line 139: What controls sharpness? 

Au: From the comments by reviewer 1, we noticed that “sharply circumscribed” can include two properties, i.e., (1) concentration difference between inflamed and noninflamed areas and (2) concentration gradient at the boundary of these areas. We now investigated what controls these properties in the model. The concentration difference depends on the parameters of the positive feedback (i.e., a, b), whereas the gradient at the boundary depends little on the parameters and the diffusion coefficient. We now added these descriptions in Results (p.8) with new Figures 2B, 2G and 3G-I.

>Line 143: What are mean? Typical based on initial condition? 

Au: Thank you for pointing out this misleading sentence. We have removed ”typical” (from p.7 line 143 previous manuscript).

>Line 149: meaning?

Au: Here, we have rewritten the relevant text (p.9 line 182-184; revised manuscript) as follows: “This series of events, that is, positive feedback of production, diffusion, and state transition in the adjacent area, occurs in each position and propagates to the surrounding noninflamed area.”

> Line 153: really? 

Au: To avoid this confusion, we now deleted the sentence (p.8 line 151-153; previous manuscript). 

>Line 155: not the reason for steep 2D doesn't show this.

Au: We have included a new Figure 2D to further show that the diameter of the inflamed area increased at a constant rate over time.

>Line 172: No, qualitatively different. D changes speed but not sign, other parameters can change the sign. 

Au: Following this comment, we have changed the relevant sentence (p.10 line 204-205; revised manuscript) as follows: “Unlike the dependence on d, the velocity continuously decreases and falls below zero at a threshold value of a and b (Figs 3B, C, and F).”

>Fig.3 caption: “(E) p=0.37, (F) p=0.5 ...”. 

Au: Following the reviewer's suggestion, we have revised the text (p.11 line 221-222; revised manuscript).

>Line 192: Explain the connection to the model and that this is an approximation. 

Au: To address your point, we now clarified the approximation in Appendix (p.16 line 343-353; revised manuscript) and Results (p.11 line 238; revised manuscript).

>Line 192: variable nor a control par. 

Au: The mediator concentration is variable. We have rewritten this sentence (p.11 line 233; revised manuscript) in line with your comments. We hope that the edited section clarifies the points we attempted to make.

>Line 193: determined <-> approximated

Au: We have corrected the term “represented” to “approximately determined” as the reviewer suggested (p.11 line 238; revised manuscript).

>Line 194: at the. 

Au: We have corrected the term “of” to “at the” as you suggested (p.11 line 238; revised manuscript).

>Line 200: not really.

Au: Based on the important comment, we re-analyzed this result using the normalized model (Eq. 3 in the revised manuscript), added new figures (S3 Fig), and revised this sentence (p.11-12 line 243-245; revised manuscript) as follows: “For the range of parameter values showing bistability, this theoretical velocity agreed with the simulated velocity in not only the sign (Figs. 3F and 4A) but also the approximate value, except for the parameter values at the velocity of zero (S3 Fig).”

>Line 204: not equal here.

Au: We have revised the relevant sentence (p.12 line 247-248; revised manuscript) as follows: “the velocity is suppressed toward zero at a similar distance from ST to SNI and to SI (Figs 4A and C).”

>Figure4 caption: other parameter is where ?

Au: Following this comment, we have added the explanation of parameter values (p.12 line 260; revised manuscript).

>Line 225: evidence - how crucial, what's normal, abnormal. 

Au: To clarify the meaning of “crucial” as the reviewer pointed out, we have changed the sentence (p.13 line 270; revised manuscript) as follows: “Erythema is characterized by the expansion of its circular area.” We have also added an explanation of the “normal” and “abnormal” (i.e. pathological) inflammatory response as follows: “In this pathological inflammatory response, the mediator level is persistently high, whereas, in a normal inflammatory response, the mediator level is temporarily elevated and then returns to its original level [12-15]” (p.13 line 270-272; revised manuscript)

> Line 226: Don’t think ref [6] is theoretical. 

Au: Thank you for correctly pointing this reference. Following this comment, we have replaced the reference [6; Zhang J-M, Int Anesthesiol Clin. 2007] with [Seirin-Lee S, PLoS Comput Biol, 2020] (p.13 line 274; revised manuscript) .　

>Line 253: other regulations not only barriers and microbiome cytokine? There is lots of published modelling of pro- and anti-inflammatory signalling and bistability, some of which should be discussed in this context.

Au: No, we do not intend the barrier and microbiome in this subsection entitled “Limitation of the present model”. We rather focus on the anti-inflammatory mediator as a possible extension of our model to account for the disappearance of erythema after the expansion. To clarify this point, we reconstructed this subsection entitled “Limitation of the present model” with adequate reference to the model studies (p.15 line 314-326; revised manuscript). For the “barriers and microbiome” the reviewer suggested, we have revised their discussion entitled “Potential application of treatments” (p.14 line 299-313; revised manuscript). 

> Line 264: This calculation is in large part directly from JD Murray’s book “Mathematical Biology”. 

Au: Following the reviewer’s comment, we have added the following sentence by explicitly referring to this book; “The previous mathematical study derived the velocity of the traveling wave in the reaction-diffusion equations [22]. We have applied this theory to Eq.(3)” (p.16 line 356-357; revised manuscript).

> S3 Fig. caption: Explain that one is for an approximation. 

Au: Following this suggestion, we have clarified the approximation at the title (Supporting Material).

---

## [Decision Letter · Decision Letter 1]

9 Nov 2021

PONE-D-21-08023R1Traveling wave of inflammatory response to regulate the expansion or shrinking of skin erythemaPLOS ONE

Dear Dr. Fujimoto,

Thank you for submitting your manuscript to PLOS ONE. After careful consideration, we feel that it has merit but does not fully meet PLOS ONE’s publication criteria as it currently stands. Therefore, we invite you to submit a revised version of the manuscript that addresses the points raised during the review process.

We look forward to receiving your revised manuscript.

Kind regards,

Daniele Avitabile

Academic Editor

PLOS ONE

Journal Requirements:

Reviewers' comments:

Reviewer's Responses to Questions

**Comments to the Author**

1. If the authors have adequately addressed your comments raised in a previous round of review and you feel that this manuscript is now acceptable for publication, you may indicate that here to bypass the “Comments to the Author” section, enter your conflict of interest statement in the “Confidential to Editor” section, and submit your "Accept" recommendation.

Reviewer #1: (No Response)

Reviewer #2: All comments have been addressed

2. Is the manuscript technically sound, and do the data support the conclusions?

Reviewer #1: Partly

Reviewer #2: Yes

3. Has the statistical analysis been performed appropriately and rigorously? 

Reviewer #1: N/A

Reviewer #2: N/A

4. Have the authors made all data underlying the findings in their manuscript fully available?

Reviewer #1: Yes

Reviewer #2: Yes

5. Is the manuscript presented in an intelligible fashion and written in standard English?

Reviewer #1: No

Reviewer #2: Yes

6. Review Comments to the Author

Reviewer #1: I appreciate that the revision addressed some of the reviewers’ comments and improved the clarity of the manuscript to some extent.

However, there are still a few scientific issues to be clarified, and too many issues with the lack of clarity in description for scientific papers. The science behind seems to be solid, but the paper writing is still poor, which is really unfortunate.

Line 14: what does “normal inflammatory response” mean?

Line 23: “the circular inflamed area expands via the traveling wave from the noninflamed to the inflamed state” in the model simulation. But it cannot be claimed as is until it is confirmed experimentally.

Line 26: “positive feedback becomes weak given the bistability” – why?

Line 28: This sentence does not make sense grammatically.

Line 28: What does “approximately controlled” mean?

Line 30: The wording “control principle” is an overstatement and should be removed from the manuscript. You cannot claim the result as a principle, unless you can demonstrate the general applicability of the “principle” to other systems. The science behind the reported work is solid, and it is sufficient to claim that the traveling wave could explain erythema expansion and shrinkage, without an overstatement which counteracts for the credibility of the paper.

Line 75: I would rather say “can cause” and “can appear” unless you can verify it experimentally.

Line 106: non-dimensionalisation?

Line 112: How does the parameter d affect? The figure only shows the case where d=0, but the choice of d also should have some effects on the existence of bistability.

Line 121: Please clarify how you define “large”.

Line 122: The relationship between models and the experimental/clinical observations is simply hypothetical. S_NI and S_I do not appear in the dermatology textbook cited there. Please make it clear that it is a speculation or a hypothesis.

Line 124: Do you assume that the high q in dermis appears as the erythema on the skin surface directly? It is a model assumption that needs to be articulated.

Line 131: How was it “confirmed”? Please demonstrate the results.

Line 156: “whether diffusion…can cause”. This paper examined whether the model could demonstrate an expansion of the erythema as a result of diffusion and bistability. It needs to be clear that the results are about model simulation. Otherwise it would become an overstatement.

Line 157: what does “consistently” mean? Always observed for different parameter values? What are the ranges investigated?

Line 160: How “large” is large?

Line 169: please clarify why you always consider the case where d=0.

Line 181: Irreversibility of the transition does not seem to have been mentioned before.

Line 248: “smaller” and “larger” than what?

Line 272: What does “persistently” mean?

Line 295: Figs 2F-J show only the cases for circumscribed erythema and do not correspond to the description of a poorly circumscribed lesion.

Line 300: Section on “Potential application of treatments” is a quite of stretch and speculative, given the theoretical analysis conducted in this paper. It may be worth removing to make the manuscript scientifically more solid.

Line 304: skin microbiome composition also affects the skin barrier integrity, and application of moisturizers also affects the skin microbiome. We cannot separate the effects of the skin microbiome and moisturizers.

Minor comment

- I would suggest reducing the use of “this” in the manuscript. It is not always clear what “this” designates for.

- Could authors share the code on GitHub?

Reviewer #2: I am satisfied with the revised version. No further comments.

I am satisfied with the revised version. No further comments.

7. PLOS authors have the option to publish the peer review history of their article (what does this mean?). If published, this will include your full peer review and any attached files.

Reviewer #1: No

Reviewer #2: **Yes: **Robert Gniadecki

---

## [Author Response · Author response to Decision Letter 1]

10 Dec 2021

Response to Reviewer #1: 

>Reviewer #1: I appreciate that the revision addressed some of the reviewers’ comments and improved the clarity of the manuscript to some extent. However, there are still a few scientific issues to be clarified, and too many issues with the lack of clarity in description for scientific papers. The science behind seems to be solid, but the paper writing is still poor, which is really unfortunate.

Authors: We thank you for providing your valuable comments on our manuscript. Following the reviewer’s suggestions, we have clarified model assumptions and results throughout the manuscript. In addition, the paper has been edited again by an experienced scientific editor, who has improved the grammar and stylistic expression of the paper. 

>Line 14: what does “normal inflammatory response” mean?

Au: Our intended meaning of “normal inflammatory response” is an inflammatory response in healthy skin, which initiates a temporal increase in the level of mediators and returns to original levels. Following the reviewer's comment, we have added this explanation to the Background (p. 3 lines 46–48; revised manuscript).

>Line 23: “the circular inflamed area expands via the traveling wave from the noninflamed to the inflamed state” in the model simulation. But it cannot be claimed as is until it is confirmed experimentally.

Au: We have added the expression “a possible mechanism in which” to clarify that this sentence (p. 2 line 24; revised manuscript) refers to the model simulation.

>Line 26: “positive feedback becomes weak given the bistability” – why?

Au: To avoid such confusion, we have removed the expression “given the bistability” from the text (p. 2 line 27; revised manuscript).

>Line 28: This sentence does not make sense grammatically.

Au: We have revised the manuscript (p. 2 line 28–30; revised manuscript), and it has been proofread again by professional English language editor.

>Line 28: What does “approximately controlled” mean?

Au: In a precise mathematical sense, the velocity of the traveling wave depends on the steady state values and nonlinearity between the steady states in the model equation [Volpert V et al. 2009]. Here we approximated that the nonlinear term is a constant (A in Appendix) and theoretically derived the velocity of the traveling wave, which is proportional to the “balance” (Eq. 5). Even under this assumption, the theoretical velocity showed a similar dependence on the model parameters (i.e., a switch between expansion and shrinkage) to the velocity in the numerical simulations (S3 Fig). Thus, we conclude that balance is a major component controlling the velocity. Following the reviewer's comment, we have changed “approximately” to “mainly” in the text (p. 2 line 29; revised manuscript).

>Line 30: The wording “control principle” is an overstatement and should be removed from the manuscript. You cannot claim the result as a principle, unless you can demonstrate the general applicability of the “principle” to other systems. The science behind the reported work is solid, and it is sufficient to claim that the traveling wave could explain erythema expansion and shrinkage, without an overstatement which counteracts for the credibility of the paper.

Au: Thank you for your accurate evaluation of our study. We have reflected this comment by replacing the “control principle” with the “experimentally testable framework” (p. 2 line 31, p. 13 line 291, p. 16 line 342; revised manuscript).

>Line 75: I would rather say “can cause” and “can appear” unless you can verify it experimentally.

Au: We have reflected this by adding “can” as suggested (p. 4 line 75; revised manuscript).

>Line 106: non-dimensionalisation?

Au: Yes. We have used the word in the sentence (p. 5 line 105; revised manuscript) following your kind suggestion: “For this purpose, we non-dimensionalized Eq. (2) by normalizing the variables and parameters as follows (See Appendix for a detailed derivation):”

>Line 112: How does the parameter d affect? The figure only shows the case where d=0, but the choice of d also should have some effects on the existence of bistability.

Au: The parameter d affects the velocity of the traveling wave (Fig. 3A); however, it does not affect the existence of bistability. The parameter region exhibiting bistability in the absence of diffusion (d = 0) agreed with that for the traveling wave in the presence of diffusion (Fig. 3F).

>Line 121: Please clarify how you define “large”.

>Line 122: The relationship between models and the experimental/clinical observations is simply hypothetical. S_NI and S_I do not appear in the dermatology textbook cited there. Please make it clear that it is a speculation or a hypothesis.

Au: We have removed the expression “large” and revised this sentence (p. 6 line 121–123; revised manuscript) without referring to the “hypothetical” relationship as follows: “Finally, as an initial condition of the model simulation (Eq. 3), we referred to the physiological condition at the onset of erythema, where one or a few small (~1mm) inflamed areas exhibited a concentration of mediators above the threshold (ST) [1].” 

>Line 124: Do you assume that the high q in the dermis appears as the erythema on the skin surface directly? It is a model assumption that needs to be articulated.

Au: Yes. Accordingly, we have articulated the assumption in the main text (p. 6 line 114–115; revised manuscript).

>Line 131: How was it “confirmed”? Please demonstrate the results.

Au: As requested, we have added the results to Fig. 2D and referred the figure (p. 7 line 133). 

>Line 156: “whether diffusion…can cause”. This paper examined whether the model could demonstrate an expansion of the erythema as a result of diffusion and bistability. It needs to be clear that the results are about model simulation. Otherwise it would become an overstatement.

Au: We have added the expression “in the model” to clarify that this sentence (p. 8 line 160; revised manuscript) refers to the model simulation.

>Line 157: what does “consistently” mean? Always observed for different parameter values? What are the ranges investigated?

Au: Here, we intend to ensure that our simulation result is consistent with clinical observations of the erythema expansion. To avoid such confusion, we have revised the text (p. 8 line 161–163; revised manuscript) as follows: “The model simulations showed that a circular inflamed area was initially caused by a transient and local perturbation to the mediator’s concentration and subsequently expanded centrifugally over time (Fig. 2A), consistently with the expansion of erythema (Fig. 1B).”

>Line 160: How “large” is large?

Au: We removed the expression “large” from the text (p. 8 line 163–165; revised manuscript) and revised the text as follows: “During the expansion, the inflamed area maintained a steep gradient of concentration at the boundary (Fig. 2B) and increased the diameter at a constant rate (velocity) over time (Figs. 2C and D).”

>Line 169: please clarify why you always consider the case where d=0.

Au: Here, we aimed to examine whether diffusion is necessary for the expansion of the inflamed area in our simulation. To clarify this, we now revised the text (p. 8 line 171–172; revised manuscript) as follows: “Without diffusion (i.e., d = 0), an inflamed area appeared; however, this area did not expand and remained constant over time (S1Fig. C).” 

>Line 181: Irreversibility of the transition does not seem to have been mentioned before.

Au: We have replaced the term “irreversible” with “selective” throughout the paper (p. 9 line 184 and line 190; revised manuscript).

>Line 248: “smaller” and “larger” than what?

Au: Following your comments, we restructured the sentence without using “larger” nor “smaller” (p. 12 line 251–253; revised manuscript) as follows: “When ST is at an equal distance from SNI and SI, given a decrease in the maximum production rate (a), the velocity is suppressed toward zero (Figs 4A and C).” 

>Line 272: What does “persistently” mean?

Au: Our intended meaning of “persistently” is that in pathological skin showing the erythema expansion, the inflammatory response persists, and the concentration of mediators fails to return to the original level. We have revised these sentences (p. 13 line 274–276; revised manuscript) to clarify the statement.

>Line 295: Figs 2F-J show only the cases for circumscribed erythema and do not correspond to the description of a poorly circumscribed lesion.

Au: Thank you for pointing out this important issue. We have summarized definitions of well- and poorly-circumscribed erythema and discussed the correspondence with our model results in a subsection titled “Expansion of a well- and poorly-circumscribed erythema” in the Discussion (p. 14 line 292–309; revised manuscript).

>Line 300: Section on “Potential application of treatments” is a quite of stretch and speculative, given the theoretical analysis conducted in this paper. It may be worth removing to make the manuscript scientifically more solid.

Au: We have removed the speculative sentences highlighted by the reviewer, and reorganized the subsection (p. 14–15 line 310–323; revised manuscript), focusing on the relevance of our mathematical model to the biological factors of the skin. 

>Line 304: skin microbiome composition also affects the skin barrier integrity, and application of moisturizers also affects the skin microbiome. We cannot separate the effects of the skin microbiome and moisturizers.

Au: We have revised the paragraph (p. 14–15 line 310–323; revised manuscript) and deleted reference [33] to focus on the probiotics without referring to the moisturizers, following the reviewer’s suggestion.

Minor comment

>- I would suggest reducing the use of “this” in the manuscript. It is not always clear what “this” designates for.

Au: We have replaced “this” with clearer terms throughout the paper.

>- Could authors share the code on GitHub?

Au: Yes. We have shared the code on GitHub, and have described the URL in main text (p. 7 line 135–136): 

https://github.com/MakiSudo/Travelingwave_Simulation/blob/bc2c10ddd5eff8db374b0804e11a63ef3c0e766a/Simulationcode.c

Response to Reviewer #2:

>Reviewer #2: I am satisfied with the revised version. No further comments.

Au: We appreciate your positive evaluation and many insightful comments on our manuscript.

---

## [Editor Report · Decision Letter 2]

12 Jan 2022

Traveling wave of inflammatory response to regulate the expansion or shrinkage of skin erythema

PONE-D-21-08023R2

Dear Dr. Fujimoto,

We’re pleased to inform you that your manuscript has been judged scientifically suitable for publication and will be formally accepted for publication once it meets all outstanding technical requirements.

Kind regards,

Daniele Avitabile

Academic Editor

PLOS ONE

---

## [Editor Report · Acceptance letter]

31 Jan 2022

PONE-D-21-08023R2 

Traveling wave of inflammatory response to regulate the expansion or shrinkage of skin erythema 

Dear Dr. Fujimoto:

I'm pleased to inform you that your manuscript has been deemed suitable for publication in PLOS ONE. Congratulations! Your manuscript is now with our production department. 

Kind regards, 

on behalf of

Dr. Daniele Avitabile 

Academic Editor

PLOS ONE